# A Comparison of Clinical Outcomes between Endoscopic Resection and Surgical Resection in Ampullary Tumors

**DOI:** 10.3390/medicina56100546

**Published:** 2020-10-18

**Authors:** Jung-Soo Pyo, Byoung Kwan Son, Hyo Young Lee, Il Hwan Oh, Kwang Hyun Chung

**Affiliations:** 1Department of Pathology, Daejeon Eulji University Hospital, Eulji University School of Medicine, Daejeon 35233, Korea; jspyo@eulji.ac.kr; 2Department of Internal Medicine, Nowon Eulji University Hospital, Eulji University School of Medicine, Seoul 01830, Korea; 2hyo0@eulji.ac.kr (H.Y.L.); 20180121@eulji.ac.kr (I.H.O.); kh.chung@eulji.ac.kr (K.H.C.)

**Keywords:** ampullary tumor, endoscopic resection, surgical resection, meta-analysis

## Abstract

*Background and objectives:* This study aimed to elucidate the clinical outcomes of endoscopic resection (ER) through comparison with surgical resection (SR) through a meta-analysis. *Materials and Methods:* This meta-analysis was performed using 32 studies. The complete resection and recurrence rates of treatment for ampullary tumors were investigated and compared between ER and SR. In addition, complications, including pancreatitis, cholangitis, cholecystitis, perforation, and papillary stenosis, and mortality of ER and SR, respectively, were estimated. *Results:* The rates of complete resection were 0.812 (95% confidence interval, CI, 0.758–0.856) and 0.929 (95% CI 0.739–0.984) in ER and SR, respectively. Recurrence rates were 0.145 (95% CI 0.107–0.193) and 0.126 (95% CI 0.057–0.257) in ER and SR, respectively. There were no significant differences in complete resection and recurrence rates between ER and SR in the meta-regression tests (*p* = 0.164 and *p* = 0.844, respectively). The estimated rates of pancreatitis, cholangitis/cholecystitis, perforation, and papillary stenosis were 12.8%, 4.4%, 5.2%, and 4.3% in ER and 9.9%, 5.6%, 2.3%, and 5.6% in SR, respectively. There was no significant difference in complications between ER and SR. The mortality rate of SR was slightly higher than that of ER (0.041, 95% CI 0.015–0.107 vs. 0.031, 95% CI 0.005–0.162). Our results show that ER had no significant differences in terms of complete resection and recurrence rates compared to SR, regardless of tumor behaviors. *Conclusions:* By comparing the complication and mortality rates between ER and SR, the safety of ER was proven.

## 1. Introduction

The ampulla of Vater is located in the junction of the common bile duct and the main pancreatic duct. Various tumors, including benign and malignant tumors, can occur in the ampulla of Vater and are infrequently compared to other gastrointestinal neoplasms [1,2,3]. Adenomas of the papilla of Vater have the potential for malignant transformation [4,5]. In addition, ampullary adenomas may be admixed with the malignant lesion in the deep portion. Although surgical resection (SR) can have the possibility of complete resection, SR has higher morbidity and mortality compared to endoscopic resection (ER) [6,7]. In previous studies, ER showed a wide range of complete resection from 9.1% to 98.4% [8,9,10,11,12,13,14,15,16,17,18,19,20,21,22,23,24,25,26,27,28,29,30,31,32]. The various factors can impact on variable rates. ERs were applied to the various lesions, including malignant tumors. The decision of appropriate treatment modality may be important in the treatment of ampullary tumors. The endoscopic biopsy from the superficial portion of the lesion cannot guarantee the concordance of histologic diagnoses between biopsied and resected specimens due to the lack of an evaluation of the deep portion. Discordance of histologic diagnoses can occur and cause inappropriate treatment and lower complete resection rates.

In the gastrointestinal tract, endoscopic submucosal dissection or mucosal resection is applied to various adenomas, especially low-grade dysplasia. However, ampullary tumors differ in anatomical and functional importance compared to other gastrointestinal tract tumors. The endoscopic approach can be appropriate with initial treatment for benign ampullary tumors rather than invasive treatment. However, a detailed comparison between ER and SR is not available in individual studies. This study aimed to elucidate the clinical outcomes of ER through comparison with surgical resection (SR).

## 2. Materials and Methods

### 2.1. Published Study Search and Selection Criteria

Relevant articles were obtained by searching the PubMed database through 15 August 2020. The search was performed using the following keywords: “Ampulla of Vater” AND “ampullectomy OR papillectomy.” The titles and abstracts of all returned articles were screened for exclusion. Review articles were also screened to find additional eligible studies. The search results were primarily included or excluded according to the following criteria: (1) studies of endoscopic or surgical resections in human ampullary tumors were included; (2) case reports or non-original articles were excluded; and (3) all articles were English-language publications. Finally, eligible studies must includ information for rates of complete resection, recurrence, and complications after the procedure.

### 2.2. Data Extraction

The following data were collected from the full texts of eligible studies: the first author’s name, study location, year of publication, procedures, information on the tumor, number of patients analyzed, rates of complete resection and recurrence, and complications after the procedure [8,9,10,11,12,13,14,15,16,17,18,19,20,21,22,23,24,25,26,27,28,29,30,31,32,33,34,35,36,37,38,39]. Any disagreements were resolved by consensus.

### 2.3. Statistical Analyses

For the meta-analysis, all data were analyzed using the Comprehensive Meta-Analysis software package (Biostat, Englewood, NJ, USA). The rates of complete resection, recurrence, and complication by procedures were investigated. In addition, comparisons of complete resection and recurrence rates between ER and SR were performed. The complications, including pancreatitis, cholangitis/cholecystitis, perforation, and papillary stenosis, were estimated using a meta-analysis. The values were pooled using a random effect model for interpretation due to performing by various operators in various populations. Heterogeneity between eligible studies was assessed using Q and I2 statistics and presented using *p*-values. In addition, the statistical difference between subgroups was confirmed through a meta-regression test. Sensitivity analysis was conducted to assess the heterogeneity of eligible studies and the impact of each study on the combined effect. To consider publication bias, Egger’s test was used. If significant publication bias was found, the fail-safe N and trim-fill tests were performed to confirm the degree of publication bias. A *p*-value of < 0.05 was considered significant.

## 3. Results

### 3.1. Selection and Characteristics of Studies

In total, 269 reports were identified in the literature search, and 32 reports were finally included in the meta-analysis. Due to non-original articles, 135 reports were excluded. Among the remaining reports, 92 were excluded due to a lack of sufficient information. In addition, 20 reports were excluded due to being articles in a language other than English (*n* = 19) or a non-human study (*n* = 1). The analyzed number of patients was 1752. In detail, 1486 and 266 patients underwent ER and SR (Figure 1), respectively. The characteristics of the eligible studies are shown in Table 1.

### 3.2. Complete Resection and Recurrence after Resection of Ampullary Tumors

First, the complete resection of ampullary tumors was estimated and compared between ER and SR. The estimated rates of complete resection were 0.812 (95% confidence interval, CI, 0.758–0.856) and 0.929 (95% CI 0.739–0.984) in ER and SR, respectively (Table 2). However, there was no significant difference of complete resection rates between ER and SR (*p* = 0.164 in a meta-regression test). In addition, the recurrence rates after ER and SR were 0.145 (95% CI 0.107–0.193) and 0.126 (95% CI 0.057–0.257), respectively. Although the recurrence rate was slightly higher in ER than in SR, there was no statistical difference in a meta-regression test (*p* = 0.844). In the primary assessment of publication bias, significant bias was found in the complete resection rate of SR (*p* = 0.018 in Egger’s test). However, there was no significant publication bias in the secondary assessment through fail-safe N and trim-fill tests.

### 3.3. Complications and Mortality Rates after Resection of Ampullary Tumors

Complications, including pancreatitis, cholangitis/cholecystitis, perforation, and papillary stenosis, were evaluated by treatment modalities (Table 3). In ER, the complication rates of pancreatitis, cholangitis/cholecystitis, perforation, and papillary stenosis were 0.128 (95% CI 0.109–0.151), 0.044 (95% CI 0.025–0.077), 0.052 (95% CI 0.038–0.071), and 0.043 (95% CI 0.027–0.068), respectively. In SR, the complication rates of pancreatitis, cholangitis/cholecystitis, perforation, and papillary stenosis were 0.099 (95% CI 0.052–0.179), 0.056 (95% CI 0.008–0.307), 0.023 (95% CI 0.001–0.277), and 0.056 (95% CI 0.008–0.307), respectively. However, there was no significant difference in complications between ER and SR. The mortality rate was slightly higher in SR compared to ER (0.041, 95% CI 0.015–0.107 vs. 0.031, 95% CI 0.005–0.162; Figure 2).

## 4. Discussion

In the current meta-analysis, the number of included patients of each study was from 3 to 161, indicating that there are fewer patients than studies in other fields. Therefore, the small number of included patients can lead to interpretation errors for the therapeutic effects. To the best of our knowledge, the present study is the first meta-analysis to compare the clinical outcomes and complications of ER and SR in ampullary tumors. 

The indications or contraindications of ER for ampullary tumors have not been established yet. The complete resection and associated complications can be affected by various factors, including tumor behaviors, tumor extent, or operator experience. These factors should be considered in deciding the treatment modality. Compared to SR, ER has the potential risk of incomplete resection [8,11,40]. As described above, the investigations for sufficient patients are needed to guarantee a comparison of the therapeutic effect or complications of ER and SR. However, the cumulative information cannot be obtained from each individual study. In addition, because some studies have various patient groups, such as benign and malignant tumors, a detailed analysis is needed. The therapeutic effect of treatment for the ampullary tumors can be evaluated by complete resection or recurrence rates. The resection margin is essential in assessing the complete resection. Evaluation of the resection margin can frequently be limited due to confounding variables and artifacts in ER. Notably, in ER specimens, the complete resection rate can be underestimated due to limitations of pathologic examination. However, handling and evaluating SR specimens is easier for resection margins and can affect complete resection rates. The evaluation of complete resection and recurrence is more difficult in ER than in SR. In the current study, although the complete resection rate was higher in the SR group than in the ER group, there was no statistical significance between SR and ER groups in the meta-regression test. 

Ampullary tumors have various extents according to tumor behaviors, such as benignity and malignancy. We evaluated the therapeutic effects of benign and malignant ampullary tumors. However, because the previously diagnosed malignant tumors cannot be treated by ER, the comparison of therapeutic effects is limited. In addition, the direct comparison of therapeutic effect for malignant ampullary tumors between ER and SR is not available in the current study. The rates of complete resection were 0.643 (95% CI 0.376–0.843) and 0.779 (95% CI 0.660–0.864) in ER for malignant and benign tumors, respectively. The therapeutic effects of ER for malignant tumors were lower than those for benign tumors. Of course, the rate of complete resection of SR was higher than that of ER for malignant tumors. However, the recurrence rate after resection of malignant ampullary tumors was higher in SR than in ER (0.259, 95% CI 0.086–0.565 vs. 0.222, 95% CI 0.009–0.902). Malignant tumors have infiltrative lesions into the stroma and cannot form a mass. That is, SR cannot guarantee the complete resection of ampullary tumors. Because SR for ampullary tumors cannot guarantee complete resection, ER for benign ampullary tumors is considered the first line of treatment rather than SR.

Each study reported the trials of various endoscopic treatments [8,9,10,11,12,13,14,15,16,17,18,19,20,21,22,23,24,25,26,27,28,29,30,31,32]. However, the results of the new trials have limitations due to the small number of patients. The different guidelines for various lesions were applied in eligible studies, and the therapeutic effects of ER and SR can be variable according to the reports [8,9,10,11,12,13,14,15,16,17,18,19,20,21,22,23,24,25,26,27,28,29,30,31,32,33,34,35,36,37,38,39]. In the current study, the overall information for the therapeutic effect and complication of ER was obtained through a meta-analysis. The results of the previous biopsy can be important in the decision of treatment modality. Based on previous studies, ER, which is less invasive, has been reported as a reliable treatment modality for benign ampullary tumors [8,9,41,42]. On the other hand, SR requires general anesthesia and laparotomy. Despite the advantage of ER, there are important considerations in deciding the treatment modality. Firstly, it is difficult to confirm the diagnosis of benign adenoma in the ampulla of Vater. Second, due to insufficient biopsy, a discrepancy of pathologic diagnoses between biopsied and resected specimens can occur. In previous references, carcinomas were detected in 25–43% of cases after resection from ampullary adenomas [41,43]. If the malignant tumor is proven in the previous biopsy, ER may be a contraindication in treatment modality. Therefore, the incidental rate of malignancy (discrepancy of pathologic diagnosis between biopsied and resected specimens) may be higher in ER than in SR. However, if endoscopic biopsied specimens are obtained from superficial mucosa, the infiltrative lesion may not be evaluated. Because of the discordant rate of diagnoses between biopsied and resected specimens of ampullary tumors, the first recommendation of SR is not reliable. The development of new instruments, including endoscopic ultrasonography (EUS), can be useful for increasing the accuracy of diagnosis in the pretreatment evaluation. The increasing accuracy of diagnosis in the previous biopsy led to the exclusion of the contraindicated cases of ER, meaning that the therapeutic effect of ER can be better. Basically, the comparison of the therapeutic effect between ER and SR may be difficult because comparison for the same lesions is impossible.

The frequency of complications can differ according to the treatment modality. We compared the complications of pancreatitis, cholangitis/cholecystitis, perforation, and papillary stenosis. In addition, the complications from only surgical resection should be considered. For example, wound infection can occur after SR, and the rate was estimated as 14.3%. Because hemorrhage may differ in amount and extent according to the treatment modality, it was not considered in the current study.

There were some limitations in the current meta-analysis. First, the recurrence rate can be affected by the follow-up period. However, the subgroup analysis based on the follow-up period could not be performed due to variable follow-up periods. Second, post-procedural pancreatitis can be increased by ER [12]. To diminish pancreatitis, pancreatic duct stents can be applied after the papillectomy [12]. However, the detailed analysis for post-procedural pancreatitis could not be performed according to the application of pancreatic duct stents. Third, the accuracy of the previous biopsy may be important in deciding the treatment modality. However, from previous studies, the low diagnostic accuracy of endoscopic biopsies might be not sufficient in differentiating between benign adenomas and carcinomas [43,44,45]. Stratification by the previous findings of the biopsied specimen is needed for comparison of complete resection between ER and SR. Fourth, in the present study, the advantages and disadvantages of surgical resection were not handled due to insufficient information. In addition, the comparisons by detailed subgroups, such as age, fitness, surgical fitness, sex, and body mass index, could not performed.

## 5. Conclusions

The complete resection and recurrence rates of ER were, respectively, 81.2% and 14.5%. Although the complete resection and recurrence of SR were respectively higher and lower than those of ER, there was no statistical difference in complete resection and recurrence rates between ER and SR. Through comparing the complication and mortality rates between EP and SA, the safety of EP was proven. Based on our results, if there is a possibility of the benign ampullary tumor, ER can be firstly considered as a treatment option. 

## Figures and Tables

**Figure 1 medicina-56-00546-f001:**
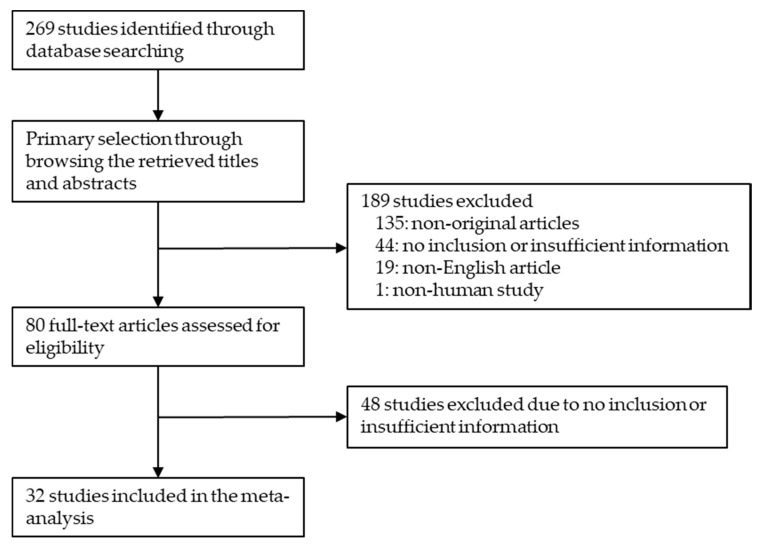
Flow chart of the study search and selection methods.

**Figure 2 medicina-56-00546-f002:**
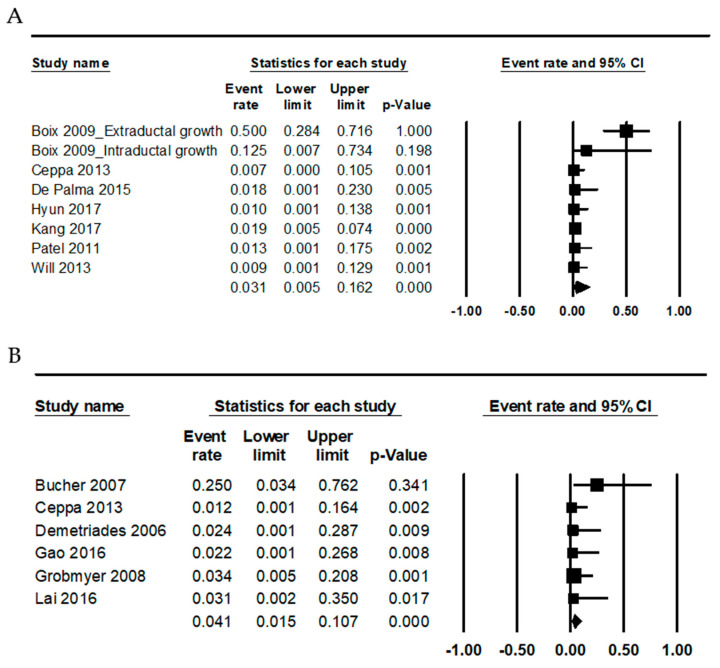
Forest plots for the estimated mortality rates in: endoscopic resection (**A**); and surgical resection (**B**).

**Table 1 medicina-56-00546-t001:** Main characteristics of the eligible studies.

First Author, Year	Location	Procedure	Subgroups	Diagnosis	No
Bohnacker 2005	Germany	EP	Extraductal growth	Mixed	75
			Intraductal growth	Mixed	31
Boix 2009	Japan	EP	Extraductal or minimal intraductal growth	Mixed	18
			Extensive intraductal growth	Mixed	3
Bucher 2007	Switzerland	SA		Adenocarcinoma	4
Ceppa 2013	USA	EA		Benign	68
		SA		Mixed	41
De Palma 2015	Italy	ESP		Mixed	27
Demetriades 2006	Greece	SA		Mixed	20
Gao 2016	China	TDA		Adenocarcinoma	22
Grobmyer 2008	USA	SA		Mixed	29
Harano 2011	Japan	EP		Mixed	28
Hong 2018	Korea	TDA	Laparoscopic TDA	Adenocarcinoma	4
			Open TDA	Adenocarcinoma	22
Hyun 2017	Korea	EP		Mixed	50
Irani 2009	USA	EP		Mixed	102
Ismail 2014	Finland	EP		Benign	35
				FAP	16
				Malignancy	10
Ito 2012	Japan	EP	with biliary and pancreatic sphincterotomy	Mixed	16
		EP	without biliary and pancreatic sphincterotomy	Mixed	12
Jeanniard-Malet 2011	France	EA		Mixed	42
Kang 2017	Korea	EP		Mixed	104
Kim 2011	Korea	EP		Benign	22
		TDA		Mixed	21
Kim 2013	Korea	EP		Mixed	57
		SA		Mixed	34
Kim 2017	Korea	TDA		Mixed	21
		EP		Benign	10
Lai 2015	Taipei	TDA		Mixed	15
Laleman 2013	Belgium	EP		Non-FAP	12
		EP		FAP	79
Lee 2016 (a)	Korea	EP	wire-guided	Mixed	22
		EP	conventional	Mixed	23
Lee 2016 (b)	Korea	TDA		Adenocarcinoma	18
Patel 2011	USA	EA		Mixed	38
Petrone 2013	Italy	ESP		Adenocarcinoma	14
Sahar 2019	USA	EP		Mixed	161
Salmi 2012	France	EA		Mixed	61
				Non-adenomatous	19
				Adenoma	29
				Malignancy	13
Soma 2015	Japan	EP		Adenoma	12
van der Wiel 2019	Netherlands	EA	confined to the papillae	Adenoma	56
			lateral spreading adenoma	Adenoma	20
			intraductal extending adenoma	Adenoma	11
Will 2013	Germany	EP		Mixed	54
Winter 2010	USA	TDA		Mixed	15
Yamamoto 2019	Japan	EA	En bloc	Mixed	125
			Piecemeal	Mixed	11

No, number of patients; EP, endoscopic papillectomy; SA, surgical ampullectomy; EA, endoscopic ampullectomy; ESP, endoscopic snare papillectomy; TDA, transduodenal ampullectomy; FAP, familial adenomatous polyposis.

**Table 2 medicina-56-00546-t002:** The estimated rates of complete resection and recurrence after treatment of ampullary tumor.

Subgroup	Number of Subsets	Fixed Effect (95% CI)	Heterogeneity Test (*p*-Value)	Random Effect (95% CI)	Egger’s Test (*p*-Value)	Meta- Regression * (*p*-Value)
Complete resection						
Endoscopic resection	25	0.796 (0.769, 0.820)	<0.001	0.812 (0.758, 0.856)	0.302	0.164
Surgical resection	4	0.841 (0.740, 0.908)	0.026	0.929 (0.739, 0.984)	0.018	
Recurrence						
Endoscopic resection	20	0.143 (0.117, 0.173)	0.006	0.145 (0.107, 0.193)	0.966	0.844
Surgical resection	9	0.157 (0.099, 0.238)	0.019	0.126 (0.057, 0.257)	0.208	

CI, Confidence interval. * Comparison between the results of endoscopic and surgical resection.

**Table 3 medicina-56-00546-t003:** The estimated rates of complications after treatment of ampullary tumor.

Subgroup	Number of Subsets	Fixed Effect (95% CI)	Heterogeneity Test (*p*-Value)	Random Effect (95% CI)	Egger’s Test (*p*-Value)	Meta- Regression * (*p*-Value)
Pancreatitis						
Endoscopic resection	31	0.127 (0.108, 0.147)	0.661	0.128 (0.109, 0.151)	0.246	0.444
Surgical resection	3	0.099 (0.052, 0.179)	0.995	0.099 (0.052, 0.179)	0.857	
Cholangitis/Cholecystitis						
Endoscopic resection	16	0.050 (0.033, 0.075)	0.066	0.044 (0.025, 0.077)	0.113	0.853
Surgical resection	1	0.056 (0.008, 0.307)	1.000	0.056 (0.008, 0.307)	-	
Perforation						
Endoscopic resection	23	0.052 (0.038, 0.071)	0.775	0.052 (0.038, 0.071)	0.038	0.552
Surgical resection	1	0.023 (0.001, 0.277)	1.000	0.023 (0.001, 0.277)	-	
Papillary stenosis						
Endoscopic resection	10	0.043 (0.027, 0.068)	0.847	0.043 (0.027, 0.068)	0.417	0.798
Surgical resection	1	0.056 (0.008, 0.307)	1.000	0.056 (0.008, 0.307)	-	

CI, Confidence interval * Comparison between the results of endoscopic and surgical resection.

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
