# Peer review of "A Comparison of Clinical Outcomes between Endoscopic Resection and Surgical Resection in Ampullary Tumors"

_medicina, 2020, doi:10.3390/medicina56100546_

Round 1

Reviewer 1 Report

Thank you.

Dear authors,

This is an interesting piece of work.

I have some comments.

1) The eternal debate between surgery and non-invasive/endoscopic surgery mirrors similar debates between surgery and other non-oncological treatment modalities.

Eg- in oesophageal cancers, the surgical treatment pathways are still very much in favour for adenocarcinoma but for squamous cell carcinoma, the evidence from painstaking clinical trials have shown that chemoradiation has equal efficacy but much less toxicity/complication risks.

2) As such, the answer of is surgery better or endoscopic treatment better cannot be answered directly from case series or even metanalysis of case series.

Surgical cases will favour younger, fitter patients with clearly abnormal investigations (scans etc) and potentially these patients will not be endoscopically treatable (may need nodal surgery etc).

Yet, if these patients do have invasive cancers, even surgery may be inadequate (alone).

Authors had already mentioned the other factor of difficulty comparing margins with ER versus enbloc SR.

3) The conclusion of 

"our results showed that ER had no significant differences of complete resection and recurrence rates compared to SR. Through comparing the complication and mortality rates between EP and SA, the safety of EP was proven" 

is not entirely certain.

The weakness of metanalysis is that even taking into account publication bias, there has not been comparison of patient factors between ER and SR.

Eg, age, fitness, surgical fitness (ASA status), gender, BMI, further treatments data are not available.

Potential confounders are that the fitter patients could undergo certain types of treatment rather than the other.

If patients are borderline operable (SR), could they have been persuaded to undergo SR rather than ER as if the margins cannot be certain, it would mean ER followed by SR?

4) " Taken together, the results show that the less invasive ER can be the first line of treatment for ampullary tumors."

If the concept is first line treatment, what is the salvage rate (SR rate) after ER?

I am not disputing that ER appears to be a better option compared to SR for patients but this statement suggests the comparison is ER+SR versus SR alone?

Otherwise, excellent work.

Thank you

Author Response

This is an interesting piece of work.

I have some comments.

1) The eternal debate between surgery and non-invasive/endoscopic surgery mirrors similar debates between surgery and other non-oncological treatment modalities.

Eg- in oesophageal cancers, the surgical treatment pathways are still very much in favour for adenocarcinoma but for squamous cell carcinoma, the evidence from painstaking clinical trials have shown that chemoradiation has equal efficacy but much less toxicity/complication risks.

Response:

Unfortunately, we did not check the detailed information for the advantages and disadvantages of surgical resections in the present meta-analysis. We added the limitation in the revised manuscript as below:

Fourth, in the present study, the advantages and disadvantages of surgical resection were not handled due to insufficient information.

2) As such, the answer of is surgery better or endoscopic treatment better cannot be answered directly from case series or even metanalysis of case series.

Surgical cases will favour younger, fitter patients with clearly abnormal investigations (scans etc) and potentially these patients will not be endoscopically treatable (may need nodal surgery etc).

Yet, if these patients do have invasive cancers, even surgery may be inadequate (alone).

Authors had already mentioned the other factor of difficulty comparing margins with ER versus enbloc SR.

Response:

Complete resection rates between benign and malignant tumors were compared in endoscopic resections (Supplementary Table 1). The rate of complete resection for malignant tumors was lower than surgical resections. We added the comment for this result in the revised manuscript as below:

The rates of complete resection were 0.643 (95% CI 0.376-0.843) and 0.779 (95% CI 0.660-0.864) in ER by malignant and benign tumors, respectively.

Of course, the rate of complete resection of SR was higher than that of ER by malignant tumors. However, the recurrence rate after resection of malignant ampullary tumors was higher in SR than in ER (0.259, 95% CI 0.086-0.565 vs. 0.222, 95% CI 0.009-0.902).

Supplementary Table 1. The estimated rates of complete resection and recurrence after treatment of ampullary tumor

Number

of

subsets

Fixed effect

[95% CI]

Heterogeneity test

[P-value]

Random effect

[95% CI]

Egger’s

Test

[P-value]

Meta-regression [P-value]

Complete resection

   Endoscopic resection

     Benign

8

0.776 [0.719, 0.824]

0.007

0.779 [0.660, 0.864]

0.993

     Malignancy

1

0.643 [0.376, 0.843]

1.000

0.643 [0.376, 0.843]

-

   Surgical resection

4

0.841 [0.740, 0.908]

0.026

0.929 [0.739, 0.984]

0.018

Recurrence

   Endoscopic resection

     Benign

9

0.156 [0.114, 0.209]

0.268

0.159 [0.111, 0.223]

0.774

     Malignancy

2

0.462 [0.241, 0.698]

0.020

0.222 [0.009, 0.902]

-

0.944†

   Surgical resection

     Malignancy

4

0.264 [0.151, 0.420]

0.063

0.259 [0.086, 0.565]

0.959

CI, Confidence interval

†, Comparison between malignancies in endoscopic and surgical resection

3) The conclusion of

"our results showed that ER had no significant differences of complete resection and recurrence rates compared to SR. Through comparing the complication and mortality rates between EP and SA, the safety of EP was proven"

is not entirely certain.

The weakness of metanalysis is that even taking into account publication bias, there has not been comparison of patient factors between ER and SR.

Eg, age, fitness, surgical fitness (ASA status), gender, BMI, further treatments data are not available.

Potential confounders are that the fitter patients could undergo certain types of treatment rather than the other.

If patients are borderline operable (SR), could they have been persuaded to undergo SR rather than ER as if the margins cannot be certain, it would mean ER followed by SR?

Response:

We agreed to the recommendation of the reviewer. However, a meta-analysis has some limitations, which cannot compare directly to two groups, as RCT. The description for this limitation was added. In addition, the conclusion was corrected as below:

In addition, the comparisons by detailed subgroups, such as age, fitness, surgical fitness, sex, and body-mass index, could not perform.

The complete resection and recurrence rates of ER were respectively 81.2% and 14.5%. Although the complete resection and recurrence of SR were prior to those of ER, there was no statistical difference in complete resection and recurrence rates between ER and SR.

4) " Taken together, the results show that the less invasive ER can be the first line of treatment for ampullary tumors."

If the concept is first line treatment, what is the salvage rate (SR rate) after ER?

I am not disputing that ER appears to be a better option compared to SR for patients but this statement suggests the comparison is ER+SR versus SR alone?

Response:

To improve, we corrected this comment in the revised manuscript as below:

Based on our results, if there is a possibility of the benign ampullary tumor, ER can be firstly considered as a treatment option.

Otherwise, excellent work.

Thank you

Reviewer 2 Report

  1. How many studies have been included in the meta-analysis? in the abstract, the authors include 31 studies, while in the methods section 32 studies are  mentioned as part of the final analysis.
  2. I believe that the inclusion criteria used by the Authors fot the study selection are too broad. For example, it should be made explicit the outcome/outcomes considered as essential to select a study for the meta-analysis 
  3. Some of the studies included in the meta-analysis are very small case series (some studies comprehend 3-4 patients). I doubt that such small case series can contribute to a meta-analysis, but rather they can lead to confounding results. I strongly suggest to exclude this studies from the meta-analysis.
  4. there are different types of endoscopic procedures described in the papers selected in the meta analysis. Do the Authors consider this a possible bias in the reported results?
  5. the selected studies deal with both benign and malignant ampullary lesions. i believe this is a potential major bias in the analysis conducted by the Authors, above all in the recurrence rate analysis. I believe that they should considered this entities separately
  6. In line 153 the Authors write "The therapeutic effects of ER for malignant tumors were lower than those for benign tumors". This statement is not clear and should be better explained by the Authors.
  7. In the discussion section there is no need to explain what a meta-analysis is.

Author Response

1.How many studies have been included in the meta-analysis? in the abstract, the authors include 31 studies, while in the methods section 32 studies are mentioned as part of the final analysis.

Response:

The number of included articles was 32. We corrected the number in the abstract.

2.I believe that the inclusion criteria used by the Authors fot the study selection are too broad. For example, it should be made explicit the outcome/outcomes considered as essential to select a study for the meta-analysis

Response:

As a recommendation, we added the detailed description of inclusion criteria in the revised manuscript as below:

The search results were primarily included or excluded according to the following criteria: (1) studies of endoscopic or surgical resections in human ampullary tumors were included; (2) case reports or non-original articles were excluded; (3) all articles were English-language publications. Finally, eligible studies should be included information for rates of complete resection and recurrence, and complications after the procedure.

3.Some of the studies included in the meta-analysis are very small case series (some studies comprehend 3-4 patients). I doubt that such small case series can contribute to a meta-analysis, but rather they can lead to confounding results. I strongly suggest to exclude this studies from the meta-analysis.

Response:

The number of cases did not include in the exclusion criteria. In addition, we performed a sensitivity analysis. These studies, which included small patients (Boix 2009; Bucher 2007; Hong 2018), had no significant impact on cumulative estimates.

4.there are different types of endoscopic procedures described in the papers selected in the meta analysis. Do the Authors consider this a possible bias in the reported results?

Response:

As pointed out by the reviewer, eligible studies obtained the findings from various populations and tumors. In addition, endoscopic resections were conducted by skilled operators using different procedures. In this case, the values were pooled using a random-effect model for interpretation due to various operators in various populations. This point is added in the method part of the revised manuscript.

The values were pooled using a random effect model for interpretation due to performing by various operators in various populations.

5.the selected studies deal with both benign and malignant ampullary lesions. i believe this is a potential major bias in the analysis conducted by the Authors, above all in the recurrence rate analysis. I believe that they should considered this entities separately

Response:

Complete resection and recurrence rates between benign and malignant tumors were compared in endoscopic resections (Supplementary Table 1). However, the detailed information of SR based on benign and malignant tumors were not included.

Supplementary Table 1. The estimated rates of complete resection and recurrence after treatment of ampullary tumor

Number

of

subsets

Fixed effect

[95% CI]

Heterogeneity test

[P-value]

Random effect

[95% CI]

Egger’s

Test

[P-value]

Meta-regression [P-value]

Complete resection

   Endoscopic resection

     Benign

8

0.776 [0.719, 0.824]

0.007

0.779 [0.660, 0.864]

0.993

     Malignancy

1

0.643 [0.376, 0.843]

1.000

0.643 [0.376, 0.843]

-

   Surgical resection

4

0.841 [0.740, 0.908]

0.026

0.929 [0.739, 0.984]

0.018

Recurrence

   Endoscopic resection

     Benign

9

0.156 [0.114, 0.209]

0.268

0.159 [0.111, 0.223]

0.774

     Malignancy

2

0.462 [0.241, 0.698]

0.020

0.222 [0.009, 0.902]

-

0.944†

   Surgical resection

9

0.157 [0.099, 0.238]

0.019

0.126 [0.057, 0.257]

0.208

     Malignancy

4

0.264 [0.151, 0.420]

0.063

0.259 [0.086, 0.565]

0.959

CI, Confidence interval

†, Comparison between malignancies in endoscopic and surgical resection

We added the comment for this result in the revised manuscript as below:

The rates of complete resection were 0.643 (95% CI 0.376-0.843) and 0.779 (95% CI 0.660-0.864) in ER by malignant and benign tumors, respectively.

Of course, the rate of complete resection of SR was higher than that of ER by malignant tumors. However, the recurrence rate after resection of malignant ampullary tumors was higher in SR than in ER (0.259, 95% CI 0.086-0.565 vs. 0.222, 95% CI 0.009-0.902).

6.In line 153 the Authors write "The therapeutic effects of ER for malignant tumors were lower than those for benign tumors". This statement is not clear and should be better explained by the Authors.

Response:

As above described, The rates of complete resection were 0.643 (95% CI 0.376-0.843) and 0.779 (95% CI 0.660-0.864) in ER by malignant and benign tumors, respectively. This comment inserted in front of that sentence as below:

The rates of complete resection were 0.643 (95% CI 0.376-0.843) and 0.779 (95% CI 0.660-0.864) in ER by malignant and benign tumors, respectively. The therapeutic effects of ER for malignant tumors were lower than those for benign tumors.

7.In the discussion section there is no need to explain what a meta-analysis is.

Response:

As a recommendation, we deleted the sentence.

Reviewer 3 Report

The authors have presented the article "A Comparison of Clinical Outcomes Between Endoscopic Resection and Surgical Resection in Ampullary Tumors" very well and no revision is required.

Author Response

The authors have presented the article "A Comparison of Clinical Outcomes Between Endoscopic Resection and Surgical Resection in Ampullary Tumors" very well and no revision is required.

Response:

Thank you for the careful review.

Round 2

Reviewer 2 Report

Dear Authors,

thank you for the revised version of the manuscript. I have only a final English correction in response number 2:

"Finally, eligible studies should be included information for rates of complete resection and recurrence, and complications after the procedure"

I would change the sentence in "Finally, eligible studies should  include data on rates of complete resection and recurrence, and complications after the procedure"

Kind regards